# E7 Oncogene HPV58 Variants Detected in Northeast Brazil: Genetic and Functional Analysis

**DOI:** 10.3390/microorganisms11081915

**Published:** 2023-07-27

**Authors:** Bárbara Simas Chagas, Elias Tibúrcio Júnior, Ruany Cristyne de Oliveira Silva, Daffany Luana dos Santos, Marconi Rego Barros Junior, Rita de Cássia Pereira de Lima, Maria da Conceição Viana Invenção, Vanessa Emanuelle Pereira Santos, Pedro Luiz França Neto, Antônio Humberto Silva Júnior, Jacinto Costa Silva Neto, Marcus Vinícius de Aragão Batista, Antonio Carlos de Freitas

**Affiliations:** 1Laboratory of Molecular Studies and Experimental Therapy (LEMTE), Department of Genetics, Federal University of Pernambuco, Recife 50670-901, PE, Brazil; babisimas@gmail.com (B.S.C.); eliasbiomedico@gmail.com (E.T.J.); ruanycristyne@hotmail.com (R.C.d.O.S.); daffanyluana94@gmail.com (D.L.d.S.); marconijrr@gmail.com (M.R.B.J.); ritajpbio@gmail.com (R.d.C.P.d.L.); maria.conceicao@ufpe.br (M.d.C.V.I.); vanessa.emanuelle@ufpe.br (V.E.P.S.); pedro.francaneto@ufpe.br (P.L.F.N.); 2Center for Biological and Health Sciences, Federal University of Campina Grande, Campina Grande 58429-900, PB, Brazil; antoniohumbertojr@yahoo.com.br; 3Laboratory of Molecular and Cytological Research, Department of Histology, Federal University of Pernambuco, Recife 50670-901, PE, Brazil; jacinto.costa@ufpe.br; 4Laboratory of Molecular Genetics and Biotechnology, Department of Biology, Federal University of Sergipe, São Cristóvão 49100-000, SE, Brazil; mbatista@academico.ufs.br

**Keywords:** human papillomavirus 58, E7 oncogene, variants, phylogenetic analysis, gene expression

## Abstract

Cervical cancer is associated with persistent infections by high-risk Human Papillomavirus (HPV) types that may have nucleotide polymorphisms and, consequently, different oncogenic potentials. Therefore, this study aimed to evaluate the genetic variability and structural effects of the E7 oncogene of HPV58 in cervical scraping samples from Brazilian women. The study was developed with patients from hospitals in the metropolitan area of Recife, PE, Brazil. The most frequent HPV types were, in descending order of abundance, HPV16, 31, and 58. Phylogenetic analysis demonstrated that the isolates were classified into sublineages A2, C1, and D2. Two positively selected mutations were found in E7: 63G and 64T. The mutations G41R, G63D, and T64A in the E7 protein reduced the stability of the protein structure. Utilizing an NF-kB reporter assay, we observed a decrease in the NK-kB pathway activity with the HPV58-E7 variant 54S compared to the WT E7. The other detected E7 HPV58 variants presented similar NF-kB pathway activity compared to the WT E7. In this study, it was possible to identify mutations that may interfere with the molecular interaction between the viral oncoproteins and host proteins.

## 1. Introduction

It is well established that cervical cancer is caused by the infection of high-risk HPVs (hrHPVs), including high carcinogenic potential genotypes [1]. Brazil’s most frequent hrHPVs are 16, 31, and 58 [2]. Together, they are the 3rd leading cause of carcinogenesis (about 16,340 new cases were reported in 2016) and the 4th leading cause of death in Brazilian women, constituting a serious public health issue. This carcinogenesis trend is similar to what is observed in other developing countries. The World Health Organization (WHO) estimated that 84% of cervical cancer cases occurred in developing countries in 2018, leading to about 311,000 female deaths. This excess death is thought to be mainly due to failure in the execution of public health programs for cervical cancer screening [3,4].

In Northeastern Brazil, HPV58 appears as one of the most frequent genotypes in cervical lesions (the second most frequent), along with HPVs 16 (1st) and 31 (3rd) [5]. HPV58 was also the second most frequent genotype in cervical intraepithelial neoplasia 3 (CIN3) lesions and the third in invasive cancer lesions [6]. As hrHPVs 16 and 31, HPV58 belongs to the *Alphapapillomavirus* genus, and plays a crucial role in cervical carcinogenesis by the expression of its oncogenes, especially E6 and E7 [7].

The impact of different genetic variants on carcinogenesis varies widely due to their influence on host immune response, cell proliferation, and the expression of specific molecules, which modify cell signaling and communication, as well as proinflammatory and apoptosis activities [8,9]. It is known, for example, that variants established by part of the Locus Control Region (LCR) region and the entire E6 gene from HPVs 16 and 18 presented higher risk for progression to CIN3 [10]. In another study, E6 and E7 oncogene variations of HPV18 have been associated with different histological forms of cervical cancer [11]. Variants produced by differences in nucleotide sequences of these genomic regions produce effects on viral replication, transcription, and the presence of specific epitopes [2,5]. Therefore, these data point to the fact that different variants have different oncogenic potentials.

Regarding HPV58, several studies have found different clinical implications for its oncogene variants, such as E7 variants. For example, a specific variant of this oncogene, T20I/G63S, was associated with an increased immortalizing potential (higher than the prototype), transforming ability, capacity of degrading pRb [12], and risk of cervical cancer due to the induced activation of AKT and K-Ras/ERK signaling pathways [9]. Moreover, evaluating gene variants is essential for identifying epitopes for developing more efficient vaccines directed to specific populations [13,14]. 

In this context, our research group has unpublished data indicating that different variants of the E7 oncogene presented different effects on NF-κB gene expression. It is well established this transcription factor plays a central role in the carcinogenesis of cervical and other cancers due to its effects on transduction signaling (e.g., Toll Like Receptors (TLRs) and Interferons (IFNs)), stromal environment, proinflammatory response, and immune surveillance [15,16].

In addition, specific variants correlate with specific clinical and epidemiological attributes, with the risk of cervical cancer and, thus, with disease prognosis [8,10,17]. Therefore, evaluating hrHPV oncogenes variants is essential to assess disease progression, establish patient prognosis and choose the best therapeutic strategy according to the clinical presentation. In this context, this study aimed to evaluate the genetic variability and structural effects of the E7 oncogene of HPV58 in cervical scraping samples from Brazilian women to identify the possible impact of these mutations in the carcinogenic process.

## 2. Materials and Methods

### 2.1. Study Groups and DNA Isolation

We enrolled 319 women with ages ranging from 18 to 77 years old, with a mean age of 37.5 (s.d. 11.9), attending the Gynecology Unit of the Clinical Hospital (*n* = 81) and the Oswaldo Cruz University Hospital (*n* = 156) at Recife, Pernambuco State, Northeastern Brazil, and the Center for Integral Attention to Women’s Health (*n* = 82) at Aracaju, Sergipe State, Northeastern Brazil. The patients came from spontaneous demand for routine examinations. Approval of the Ethical Committee (CEP/CCS/UFPE N° 49/11) and informed consent from all patients enrolled in the study were obtained.

The samples came from cervical scraping, collected using a gynecological brush (cytobrush) that reaches the endocervix. The brush containing the collected gynecological material was transferred to a sterile tube containing PBS solution (pH 7.4) to preserve the cells until DNA extraction. Cervical cells were stored at −80 °C, and the DNA was extracted using the DNeasy Blood Tissue Kit (Qiagen®—Hilden, Germany) according to the manufacturer manual. 

### 2.2. HPV Detection and Genotyping

The human MDM2 gene was amplified by Polymerase Chain Reaction (PCR) (forward primer: 5′-GATTTCGGACGGCTCTCGCGGC-3′; reverse primer: 5′-GATTTCGGACGGCTCTCGCGGC-3′) to evaluate the quality of DNA, avoiding false negative results [18]. The PCR reaction was performed in 20 µL volume, containing 2 µL (50 ng/µL) of the template DNA, 10 µL of GoTaq Green PCR Master Mix (Promega®—Madison, WI, USA), 1.2 µL of each primer (12.2 pM/µL) and 5.6 µL of Nuclease-Free Water (Promega). The reaction condition was 95 °C for 5 min, followed by 35 cycles of 94 °C for 30 s, 66.5 °C for 40 s, 72 °C for 45 s, and then 72 °C for 10 min. 

HPV DNA was detected by PCR based on the amplification of a fragment of the L1 gene using the consensus primers MY09 (5′-CGTCCMARRGGAWACTGATC-3′) and MY11 (5′-GCMCAGGGWCATAAYAATGG-3′) [19]. The PCR reaction was performed in 25 µL volume, containing 2 µL (50 ng/µL) of the template DNA, 12.5 µL of GoTaq Green PCR Master Mix (Promega®—Madison, USA), 1.5 µL of each primer (10 pM/µL) and 7.5 µL of Nuclease-Free Water (Promega®—Madison, USA). The reaction condition was 94 °C for 1 min, followed by 35 cycles of 94 °C for 30 s, 55 °C, 72 °C for 1 min, and then 72 °C for 10 min. 

HPV genotyping was also performed by PCR using specific primers for HPV-58 (forward primer: 5′-CCGTTTTGGGTCACATTGTTCATGT-3′ located at positions 7781–7805; reverse primer: 5′-AAGCCTATTTCATCCTCGTCTGAG-3′ located at positions 666–689), HPV16 (forward primer: 5′-TTCTAAGGCCAACTAAATGTCACCC-3′ located at positions 7753–7778; reverse primer: 5′-TAGAGATCAGTTGTCTCTGGTTGC-3′ located at positions 623–647) and HPV31 (forward primer: 5′-CGTTTTCGGTTACAGTTTTACAAGC-3′ located at positions 7846–7871; reverse primer: 5′-AGCTGGACTGTCTATGACAT-3′ located at positions 665–685). PCR was performed in 25 µL volume, containing 2 µL (50 ng/µL) of the template DNA, 12.5 µL of GoTaq Green PCR Master Mix, (Promega®—Madison, USA), 1.5 µL of each primer (10 pM/µL) and 7.5 µL of Nuclease-Free Water (Promega®—Madison, USA). The reaction condition was 94 °C for 2 min, followed by 35 cycles of 94 °C for 10 s, 59 °C (HPV58), 57 °C (HPV16) and 54 °C (HPV31) for 20 s, 72 °C for 1 min, and then 72 °C for 5 min. No template (milli-Q water) negative control was used in all reactions. The PCR products were examined on the 2% agarose gel stained with ethidium bromide.

All HPV58 positive samples were tested for E7 genetic variability using specific primers (E7 forward primer: 5′-ATTTGTCAAAGACAATTGTGTCCAC-3′ located at positions 488–509, E7 reverse primer: 5′-TTAAATCTGTACCACTATCGTCTGC-3′ located at positions 1000–1024). PCR was performed in a final volume of 50 µL, containing 4 µL of the template DNA (50 ng/µL), 2 mM MgSO4 (Invitrogen®—Waltham, MA, USA), 0.2 mM dNTP mix (Promega®—Madison, USA), 10 pM of each primer, and 1U Platinum Taq DNA polymerase High Fidelity (Invitrogen®—Waltham, USA). The reaction condition was 94 °C for 2 min, followed by 35 cycles of 94 °C for 10 s, 58 °C for 20 s, 72 °C for 1 min, and then 72 °C for 5 min. The sequencing of the positive samples was carried out (both strands) using ABI PRISM BigDye Terminator Cycle Sequencing V.3.1 kit (Applied Biosystems®—Waltham, MA, USA), with the exact reverse and forward primers used in the amplification reaction. The sequences were checked for quality and assembled using the Staden package [20]. HPV58 E7 sequences were compared to other HPV sequences from GenBank using Basic Local Alignment Search Tool (BLAST) [21]. Subsequently, the DNA sequences from this study were aligned with the HPV58 prototype sequence (GenBank: D90400.1) using MEGA (version 7.0) [22].

### 2.3. Phylogenetic Analysis

A Maximum Likelihood phylogenetic tree was reconstructed based on the sequences of the E7 gene by using MEGA11 [23]. The evolutionary model that best fits the data was used (K80 + I), which was selected based on the Bayesian Information Criteria (BIC) using jModelTest 2 [24]. SPR was used as ML heuristic method and moderate branch swap filter. Branch support was assessed with 1000 bootstrap replicates.

The sequences were phylogenetically classified as lineages A, B, C, and D, and sublineages A1, A2, A3, B1, B2, C1, D1, and D2. The reference genomes used for this classification were retrieved from PaVE database: D90400 (A1), HQ537752 (A2), HQ537758 (A3), HQ537762 (B1), HQ537764 (B2), HQ537774 (C1), HQ537768 (D1) and HQ537770 (D2). Other HPV58 isolates with complete genomes were obtained from the NCBI Variant Search link, which was available at the PaVE database (https://pave.niaid.nih.gov/explore/variants/variant_searches, accessed on 20 May 2022), and they were used in the phylogenetic analysis and displayed in the tree with their GenBank Accession Number.

### 2.4. Analysis of Selective Pressure

Estimates of Maximum likelihood were analyzed using the CODEML program, incorporated into PAML version 4.9c, to evaluate the selective pressure that might affect the mutations in the HPV58 E7 oncogene [25]. The detection of positive selection was performed by calculating six codon substitution model parameters, M0, M1, M2, M3, M7, and M8, which use ω = dN/dS. The basic model uses the ω = dN/dS, which means the ratio of non-synonymous/synonymous substitution rates. The branch models allow the ω ratio to vary among branches in the phylogeny and help detect positive selection acting on particular lineages [26,27]. In addition, the site models allow the ω ratio to vary among sites (among codons or amino acids in the protein) [27,28]. The Likelihood Ratio Test (LRT) was used to evaluate which model best fits the data.

### 2.5. Protein Structure Modeling

The 3D structure of the HPV58 reference and mutated E7 proteins were in silico modeled. The amino acid sequence of the HPV58 E7 reference protein (BAA31846) was used to determine its predicted 3D structure. A blast search was carried out in Protein Data Bank (PDB) using the Blastp algorithm for template selection. The template search for HPV58 E7 protein demonstrated only the C-terminal of the protein (PDB id: 2F8B) in PDB.

Therefore, the structure model of HPV58 E7 protein was predicted using a combined comparative and ab initio method in the Robetta server [29]. The models were refined by energy minimization using the 3Drefine server [30]. The predicted models were evaluated by PROCHECK [31], MolProbity [32], QMEAN [33], and TM-align [34]. According to these parameters, the best model was chosen to assess the structural variability.

The non-synonymous mutations identified were inserted on the HPV58 E7 reference structure using the Point Mutation service in the RosettaBackrub server [35]. The mutated models with the lowest weighted score were selected. The Site Directed Mutator (SDM) method [36] was used to predict the effect of the mutations on the HPV58 E7 protein structure stability.

### 2.6. T-Cell and B-Cell Epitope Prediction

The prediction of T and B cell epitopes in the HPV58 E7 protein sequences was performed using the ProPred-1 server (MHC class-I) [37], the ProPred server (MHC class-II) [38], and the ABCpred server [39], respectively.

### 2.7. Plasmid Constructs

After the analysis of the variability in the E7 oncogene sequences (297 bp), three isolates called HPV58-E7 variant 54S (HPV58/UFPE-54S), HPV58-E7 variant 58S (HPV58/UFPE-58S), and HPV58-E7 variant 60M (HPV58/UFPE-60M) were chosen because they are more epidemiologically relevant, and they present more non-synonymous alterations, belonging to lineages A, C and D, respectively. These isolates/variants and the prototype (reference gene sequence without changes) of HPV58 E7 were amplified by PCR and cloned into a vector for PCR products (pGEM-T easy—Promega), following the manufacturer’s instructions. Subsequently, they were subcloned into an empty vector (pCDNA 3.1 (+)), which is a mammalian cell expression vector. Clones were subjected to automated sequencing using the ABI PRISM BigDyeTM Terminator Cycle Sequencing Kit v3.1 Ready Reaction (Applied Biosystems^®^—Waltham, USA) in an ABI Prism 3100 automated DNA sequencer (Applied Biosystem^®^—Waltham, USA).

### 2.8. Transfection of the C33A Cells with Recombinant Expression Vectors

After being confirmed by sequencing, the recombinant vectors containing the variant sequences of the HPV58 E7 variants and prototype were isolated by maxi-preparation using the Plasmid Plus Maxi kit (Qiagen®—Hilden, Germany), according to instructions. Keratinocytes resulting from cervical carcinoma were co-transfected with an NF-κB reporter and constructs encoding the HPV58 E7 variants to evaluate the activity of the E7 oncogene, using the NF-κB pathway as a model of functionality through luminescence C33A cells. Transfections were performed in 6-well cell culture plates, where 5 × 10^5^ cells were plated in 3 ml of culture of Dulbecco’s ModifiedEagle’s Medium (DMEM-Invitrogen^®^—Waltham, MA, USA) plus 10% fetal bovine serum (Gibco^®^—Waltham, MA, USA); 1% L-Glutamine (Sigma^®^— St. Louis, MO, USA)—Complete DMEM. Cells were transfected with the recombinant vectors using the Polyfect transfection reagent (Qiagen®—Hilden, Germany). The groups of cells were transfected using the pcDNA3.1 (+) plasmid containing 1.5 µg of the E7 variant, the prototype (reference gene without any of the studied variations), or with the empty pcDNA3.1 (+) vector (negative control). The NF-kB-dependent firefly luciferase reporter used for the co-transfections was constructed with three kB factor binding sites named (kB) 3-Luc. Consequently, 1 µg NF-kB reporter was used per co-transfection, along with 1 ng of a Renilla luciferase-expressing plasmid to normalize luminescence values. After 24 h of transfection, TNFα (10 ng/mL) was added to the plates for 6 h to stimulate the pathway. All experiments were performed in three experimental replicates in triplicates. The Dual-Luciferase Reporter Assay System (Promega®—Madison, USA) was used to measure the luminescence according to the manufacturer’s instructions. Relative luciferase units (RLUs) readings were performed on the GloMax^®^ 96 Microplate Luminometer w/Dual Injectors (Promega®—Madison, USA).

### 2.9. Evaluation of the E7 Gene Expression in Transfected C33A Cells

Total RNA was obtained from C33A cells transfected with the expression vectors for the E7 variants using the RNeasy mini kit (Qiagen®—Hilden, Germany). According to the manufacturer’s instructions, complementary DNA (cDNA) was synthesized using the Improm^®^ Reverse Transcription Kit (Promega®—Madison, USA). Quantitative Polimerase Chain Reaction (qPCR) was performed using the QuantiTect SYBR Green^®^ PCR kit (Qiagen®—Hilden, Germany), and the concentrations of primers and cDNA were the same as those described by LEITÃO et al. 2014. Gene expression was normalized using the reference genes glyceraldehyde-3-phosphate dehydrogenase (GAPDH) and actin beta (ACTB). The protocol for qPCR was: (i) denaturation at 95 °C for 5 min; (ii) 40 cycles of denaturation at 95 °C for 10 s, annealing at 55 °C for 30 s (reference genes) or at 60 °C (E7 prototype and variants), extension at 72 °C for 30 s; (iii) extension at 72 °C for 10 min. Each reaction was performed in three biological replicates, including a negative control for each gene. Gene expression was calculated using [40].

### 2.10. Statistical Analysis 

Statistical analysis was performed using a one-way analysis of variance (ANOVA) followed by Bonferroni correction post-test using GraphPad Prism v.7 software. *p* < 0.05 was considered significant.

## 3. Results

### 3.1. HPV Detection and Typing

Among the 319 collected samples, 227 (71.2%) tested positive for HPV. The genotyping revealed that 55 (24.2%) samples tested positive for HPV16, 36 (15.9%) tested positive for HPV31, and 18 (7.9%) tested positive for HPV58. Considering the 227 HPV-positive samples, 59 samples present a coinfection with more than one HPV type: HPV16 + HPV31 was the most frequent (48, 20.8%), followed by HPV16 + HPV58 (6, 2.6%) and HPV16 + HPV31 + HPV58 (5, 2.2%).

HPV58 samples were identified as HPV58-E7 variant 58S, -30S, -31S, -54S, -88S, -60M, -59M, -1M, -29M, -35P, -45M, -79M, -92M, -28M, -28S and -48S. The pathological diagnosis was obtained for these samples, as specified in Table 1. Only samples with a single HPV58 infection were selected for analysis. Two samples positive for HPV58 were excluded due to the absence of a conclusive pathological diagnosis.

### 3.2. HPV58 E7 Sequence Variations

All E7 variants demonstrated nucleotide changes, and some of them were non-synonymous. The HPV58-E7 variant 54S was pathologically classified as HSIL, and the other variants as LSIL.

Nucleotide sequences of the complete E7 oncogene were compared with the HPV58 reference sequence (D90400.1). The comparative analysis of the E7 oncogene revealed that all 16 sequences (100%) presented 10 variable sites. Five mutations happened at codons 41 (G41R), 63 (G63D), 64 (T64A), 74 (T74A), and 76 (D76E) (Table 1). The remaining five mutations were silent (Table 1). The sites that presented the most frequent mutations were 744 (100%, 16 of 16), 761 (93.8%, 15 of 16), and 694 (81.3%, 13 of 16), respectively. The silent mutations were T756C (6.3%, 1 of 16), A763G (12.5%, 2 of 16), A793G (18.8%, 3 of 16), C798T (18.8%, 3 of 16), C801A (18.8%, 3 of 16), C840T (18.8%, 3 of 16), and T852C (6.3%, 1 of 16). Nucleotide substitutions resulting in premature stop codons or frameshift changes were not detected.

### 3.3. Phylogenetic Analysis

The evolutionary relationships among HPV58 E7 sequences were analyzed. The phylogenetic tree has demonstrated a lineage and sublineage-based clustering of the isolates, which groups HPV58 E7 variants into four lineages and eight sublineages (Figure 1). The phylogenetic tree presents well-supported clusters, making the HPV58 E7 variants’ genotyping possible. Thirteen isolates could be classified as sublineage A2, one as sublineage C1, and two as sublineage D2.

### 3.4. Selective Pressure Analysis

The genomic region that includes the E7 gene of HPV58 presented ω ratios less than 1, which indicates the purifying selection and acts to conserve the protein function. The model that best fits the data is the one that demonstrates higher values of log-likelihood (lnL). For this genomic region, the best model is the M3, with ω = 0.87940 (Appendix A). Although the genomic region E7 of HPV58 presented a negative or purifying selection, the LRT test indicated the presence of positively (diversifying) selected protein sites, representing mutations with a high probability of being fixed in the viral population (Table 2). Five positively selected sites were identified: 41G, 63G, 64T, 74T, and 76D. Of these, four sites (41G, 63G, 74T, and 76D) were identified to be positively selected with statistical significance (*p* > 95% and *p* > 99%) (Table 2).

### 3.5. Structural Analysis of E7 Mutations

The E7 protein was modeled using a hybrid method, with the C-terminal with a comparative method and the A-terminal with an ab initio method. The best HPV58 E7 protein model presented 92.9% residues in the most favored regions in the Ramachandran plot, with a MolProbity Score of 0.79, QMEAN score of −1.16, and TM-align score of 0.61714, which represents a well-resolved structure model. All of these quality parameters confirm the reliability of the protein structure model.

The HPV58 E7 structure has 98 amino acid residues, with one zinc-binding domain consisting of two CXXC motifs separated by 29 amino acid residues. The protein structure presents four alpha-helix domains and two beta-sheet domains connected with loop regions (Figure 2). The motif 22LXCXE26 consists of a part of an alpha helix and a loop region. No non-synonymous mutations found in this study directly affect these motifs. However, two non-synonymous mutations (G63D and T64A) are located in the loop region right after one CXXC motif (Figure 2). The mutation G41R is located in the second alpha helix region, and the mutations T74A and D76E are located in the third alpha helix. Evaluating the impact of these mutations on the protein stability for the HPV58 E7 protein, we observed that the mutations G41R, G63D, and T64A presented negative predicted values of ΔΔG, leading to protein instability and malfunction (Table 3).

### 3.6. T-Cell and B-Cell Epitopes Prediction

We performed epitope predictions to evaluate the impact of these mutations on the immune function of the E7 transforming protein. We observed amino acid changes in sites belonging to T-cell (MHC class-I and class-II) and/or B-cell epitopes, as reported in Figure 3, which shows the consensus of T-cell and B-cell epitopes obtained for E7 oncoprotein.

### 3.7. Effect of E7 Polymorphic Variants on the NF-kB Pathway

The functional study of the HPV58 E7 oncogene variants was developed using luminescence assays with the NF-kB pathway as a model. Initially, the effect on the NF-kB activity of the E7 prototype, both HPV16 and HPV58, was evaluated (Figure 4A,B). The results demonstrated that the E7 prototype of HPV16 reduced NF-kB activity. The action of each variant on NF-kB activity was also verified. The results demonstrated that all variants induced NF-kB compared to the empty vector. However, the 58S and 60M variants demonstrated similar NF-kB induction compared to the E7 prototype of HPV58 (Figure 4C). The difference between the means of the analyzed groups was significant, with *p*-value 0.0046. However, no statistically significant result was found when comparing the prototype with each variant. The comparisons between pcDNA and the prototype and pcDNA and the HPV58-E7 variant 54S were significant, with *p*-value 0.001 for all comparisons.

### 3.8. E7 Oncoprotein Gene Expression in C33A-Transfected Cells with HPV58 E7 Variants

We hypothesized that the reduction of NF-kB activity caused by the E7-54S variant might be due to decreased levels of E7 mRNA. Therefore, we evaluated the mRNA levels of the different variants (Figure 4D). The results demonstrated the relative expression levels of the HPV58/E7-prototype. However, the relative expression level of the 54S variant was significantly lower than in the HPV58/E7-prototype. Therefore, this result indicates that the low NF-kB activity presented by the HPV58/E7 variant 54S may be due to low mRNA availability.

## 4. Discussion

Cervical cancer is a severe public health issue, and it is strongly linked to high-risk HPV infections [1]. In Brazil, HPVs 16, 31, and 58 are the most prevalent high-risk genotypes [2]. E7 is considered one of the most important oncogenes, and nucleotide variations in this oncogene play a crucial function in cervical cancer due to their ability to induce cellular transformation and immortalization [41]. Studies have investigated HPV16 E7 variants [42,43]. However, the HPV58 E7 variant, considered a high-risk oncogenic type, has been scarcely studied, mainly in Brazil.

Therefore, this study has investigated sequence variations of the HPV58 E7 oncogene in cervical scraping samples from Brazilian women to evaluate their structural and functional effects in order to identify the possible impact of these mutations in the carcinogenic process. In this context, nucleotide changes previously described in the HPV58 E7 oncogene were detected at positions G694A [44,45,46], T744G [43,44,45,47], T756C [43], G761A [43,44,45,47], A763G [7], A793G, C798T, C801A, C840T and T852C [44,48].

From the phylogenetic point of view, HPV lineages have been associated with different risks for cervical cancer development, as observed for some HPV16 lineages [49,50,51]. Concerning HPV58 phylogeny, sequence variants are associated with the risk for CIN3 (cervical intraepithelial neoplasia grade 3) and invasive cervical cancer [7]. HPV58 consists of four variant lineages (A, B, C, and D) and eight sublineages (A1, A2, A3, B1, B2, C, D1, and D2) [52]. In this study, our analysis of the E7 sequences of HPV58 variants identified sublineages A2 (the majority), C1, and D2. It is essential to point out that the lineage determination of HPV relies on complete genome sequences, and a possible bias might occur when using only the E7 gene sequences. Therefore, to confirm the lineage classification in this study, a complete genome sequence comparison and phylogenetic analysis were performed with the reference HPV58 complete genome sequences. Once the lineage classification was confirmed for the reference genome sequences, the phylogenetic analysis performed with the E7 gene sequences (Figure 1) demonstrated a similar topology, indicating the isolates’ probable lineage.

In the study by Chan et al. [53], the association between oncogenic risk and variant lineage was analyzed. Lineage A represented the majority of HSIL/carcinoma samples. However, there was no statistically significant association [53]. In the present study, most of the A2 line is represented by HSIL samples. Nevertheless, further studies are necessary to evaluate the association between the variant lineages and cervical cancer risk.

The analysis indicated that the variant E7 gene of HPV58 presented a negative or purifying selection. However, evolutionary diversifying selected amino acid residues have been identified in this study. The diversifying selection is associated with mutations with a high probability of being fixed in the viral population, being related to the host adaptive response [54]. Four non-synonymous mutations were identified under the diversifying selection in the E7 protein: 41G, 63G, 74T, and 76D. These mutations were consistently found in other studies worldwide, highlighting the evolutionary relevance for the virus [45,55,56]. Although more experimental evidence is needed, these findings are relevant because these mutations might be related to the adaptation to the human host, increasing the ability of these oncoproteins to interact with the host proteins.

Several studies have been carried out investigating variants of HPV16-E5, for example [57,58,59], while variants of HPV58 E7 are poorly studied, although HPV58 is part of the high-risk group of HPVs and is a very prevalent type in some countries of the world, including some regions of Brazil. In the study by Silva et al. [60], which evaluated the activity of the NF-kB pathway mediated by variants of the HPV31 E5 gene, it was reported that all variants increased the activity of the pathway. Nucleotide variations present in the HPV-E7 oncogene play an important role in the development of cervical cancer. This is due to its ability to inactivate products of important tumor suppressor genes, such as pRb, and the malignant transformation of virus-infected cells [61]. The NF-kB pathway and its activation are frequent events in squamous cell carcinoma and provides important evidence of epithelial cell modification [62]. It is widely known that the E7 protein interacts with various biological pathways and plays an important role in oncogenesis [63,64,65]. Therefore, genetic variants of E7 may impact these pathways by altering protein–protein interaction rather than affecting their stability [66,67]. Some studies demonstrated that E7 oncogene variants suppressed the activity of the NF-kB pathway [68,69]. The HPV16-E7 protein interferes with NF-κB signaling, interacting directly with the IKKα and IKKβ subunits, causing a reduction in the activity related to NF-κB signaling in U2OS cells [69]. Even under TNF-α cytokine stimulation, U2OS cells transfected with the HPV16-E7 gene were less responsive, exhibiting attenuated NF-κB signaling [70]. 

In this study, contrary to the HPV16-E7 prototype, the HPV58-E7 prototype and its variants activated the pathway compared to an empty vector. In this scenario, we suggest that the HPV58-E7 probably acts in the NF-kB via modulating the physiology of infected cells, and consequently, reflecting in the different carcinogenesis process compared to the HPV16-E7. However, other studies are still needed to confirm this hypothesis.

Albeit all the tested variants activate the NF-kB pathways; we observed that the HPV58-E7 variant 54S seems to decrease the level of activation in comparison to the prototype and the other tested variants. This difference is likely due to the lower E7 expression observed in the HPV58-E7 variant compared to the prototype and the other tested variants. Although we did not test the variant sequences’ stability, bioavailability, or decay rate, this reduced expression level may be attributed to some of these mechanisms [71,72,73].

Furthermore, in the structural analysis for the E7 protein, the LXCXE motif is known to be related to the interaction of E7 with pRB. However, the non-synonymous mutations detected in this study did not alter the LXCXE motif. Two mutations (G63D and T64A) are located in the loop right after the CXXC motif, an important zinc-binding motif. G63D has been reported to have a significantly lower risk associated with cervical neoplasia [7]. Other mutations detected in this study (G41R, T74A, and D76E) are in the second and third alpha-helix. In a previous study, G63D and G41R have been reported to demonstrate a significantly lower risk associated with cervical neoplasia [7].

Moreover, the mutations G41R, G63D, and T64A in the E7 protein were predicted to lead to protein instability and malfunction. Interestingly, the variant with decreased NF-kB activation (HPV58/UFPE-54P) exhibits two of these mutations that lead to decreased stability (G41R and G63D). To the best of our knowledge, this is the first time these mutations have been structurally characterized. However, the impact of these mutations on the interaction of these HPV oncoproteins and host molecules still needs to be assessed in further experimental studies.

The impact of NF-kB on regulating the inflammatory process has been extensively studied, but its role in the carcinogenic process, particularly in HPV+ tumors, still requires further efforts [74]. However, it has already been demonstrated that modulating this pathway impacts the clinical outcome of HPV+ head and neck cancer patients [75]. Additionally, Cai et al. demonstrated that NF-kB is critical in tumor progression in HPV+ cervical cancer mediated by PD-L1 [76]. Therefore, our data help to reinforce the potential use of the NF-kB pathway as one of the modulators of HPV-mediated carcinogenesis and its possible utilization as a biomarker of clinical outcome. 

## 5. Conclusions

Our study confirmed the presence, in our cohort, of previously described HPV58 variants, which induce NF-kB activity, contrary to the reported effect of HPV16-E7. We also observed that the different HPV58 variants had a distinct profile of NF-kB induction. The variants present in HPV58-E7 suggest that different interactions of E7 with target proteins may occur in infected cells. Since some of the alterations occur in epitope regions, it is suggestive that this may influence the recognition of infected cells. Our results highlight the need for further in-depth studies on HPV58 and its variants. They also indicate the potential of these variants to modulate the pathophysiology of infected cells in a distinct manner, which may impact carcinogenesis. Although our study focused on HPV58-E7, it would be interesting to evaluate other oncoproteins as well.

## Figures and Tables

**Figure 1 microorganisms-11-01915-f001:**
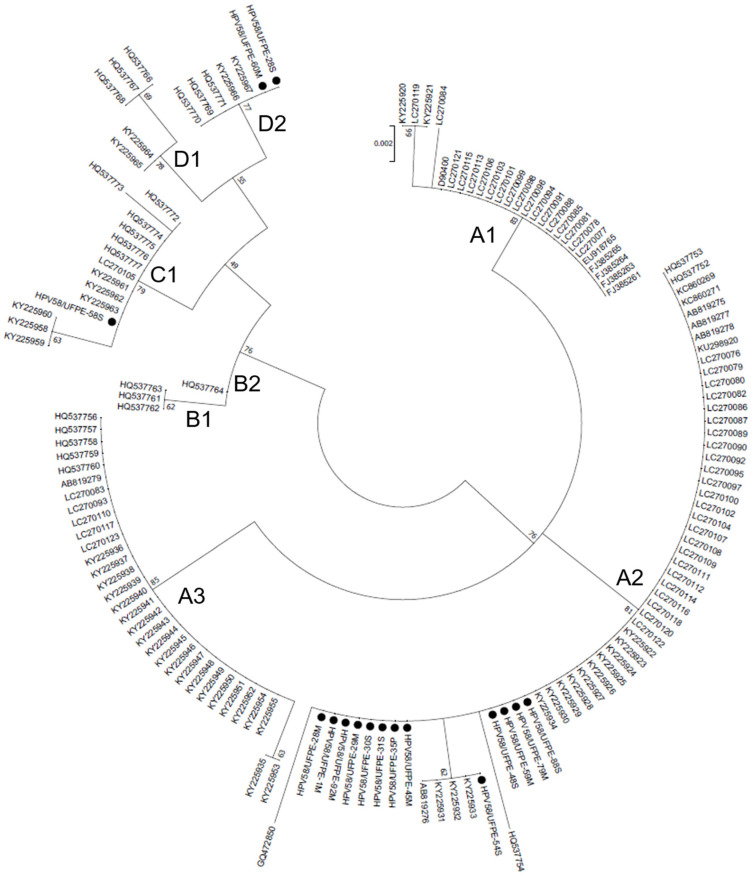
Midpoint-rooted Maximum Likelihood phylogenetic tree of HPV58 strains based on E7 gene sequences. Four clusters were identified as lineages A, B, C, and D, and eight sub-clusters were identified as sublineages. Bootstrap replicates are represented as branch support.

**Figure 2 microorganisms-11-01915-f002:**
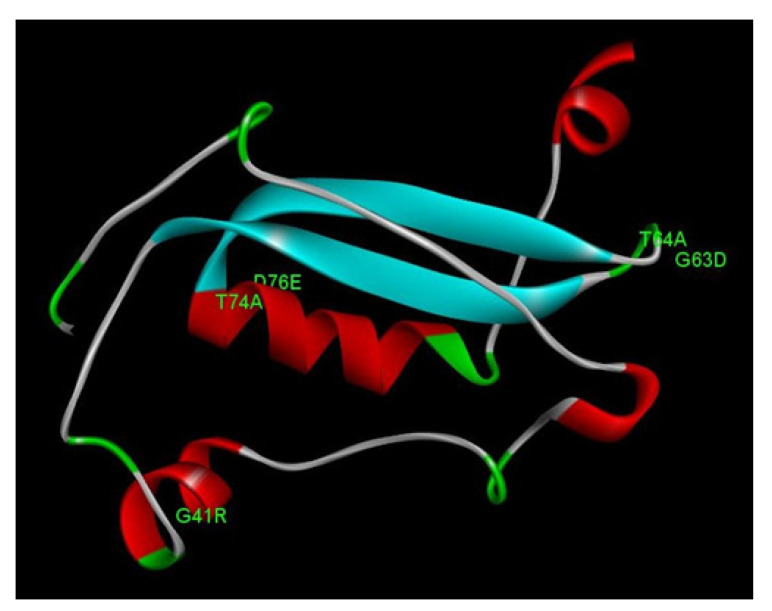
Representation of HPV58 E7 protein model as displayed in Discovery Studio Visualizer. Non-synonymous mutations found in this study are represented in the structure. Four alpha helices are represented in red, beta sheets in blue, and loops in gray and green.

**Figure 3 microorganisms-11-01915-f003:**
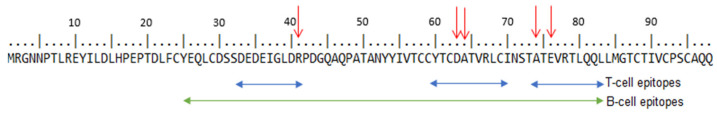
Predicted T-cell and B-cell epitopes. Epitopes within the E7 oncoprotein of HPV58 variants. In blue is a consensus of the predicted T-cell epitopes. In green is a consensus of the predicted B-cell epitopes. Red arrows indicate mutation sites presented in E7 HPV58 isolates.

**Figure 4 microorganisms-11-01915-f004:**
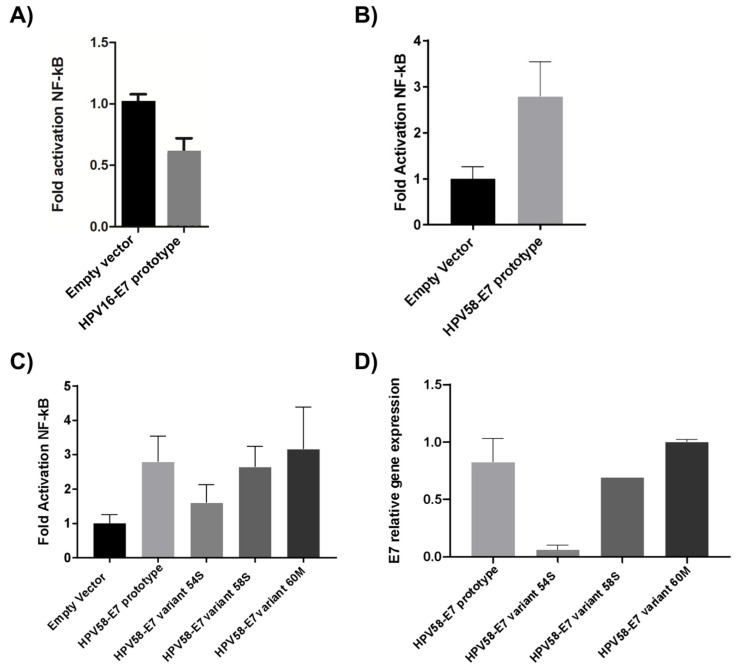
Effect of the E7 oncoprotein from HPV16 (**A**) and HPV58 (**B**) on NF-kB activity. The E7 oncoprotein from HPV16 reduces NF-kB activity, while the E7 from HPV58 increases it. (**C**) Effect of the HPV58 prototype and variants on NF-kB activity. The 58S and 60M variants do not exhibit a statistically significant difference compared to the prototype. However, the 54S variant, while lacking statistical significance, appears to demonstrate a lower activity in the NF-kB pathway. (**D**) Comparison of gene expression among different HPV58 variants and the prototype. While the 58S and 60M variants did not exhibit a significant difference in expression compared to the prototype, the 54S variant demonstrated a notably reduced expression level.

**Table 1 microorganisms-11-01915-t001:** Pathological diagnosis of HPV58 positive samples with nucleotide and amino acid variability of HPV58 E7 gene and the putative biological function affected. Conservative nucleotides concerning the reference sequence (D90400.1) are marked with a dot (.). Reference: prototype sequence of HPV58.

	E7									
Nt	694	744	756	761	763	793	798	801	840	852
Reference	G	T	T	G	A	A	C	C	C	T
*LSIL*										
HPV58-E7 58S	.	G	.	.	.	G	T	A	T	C
HPV58-E7 88S	A	G	.	A	.	.	.	.	.	.
HPV58-E7 60M	.	G	.	A	G	G	T	A	T	.
HPV58-E7 28M	A	G	.	A	.	.	.	.	.	.
*HSIL*										
HPV58-E7 30S	A	G	.	A	.	.	.	.	.	.
HPV58-E7 31S	A	G	.	A	.	.	.	.	.	.
HPV58-E7 54S	A	G	C	A	.	.	.	.	.	.
HPV58-E7 59M	A	G	.	A	.	.	.	.	.	.
HPV58-E7 1M	A	G	.	A	.	.	.	.	.	.
HPV58-E7 45M	A	G	.	A	.	.	.	.	.	.
HPV58-E7 79M	A	G	.	A	.	.	.	.	.	.
HPV58-E7 92M	A	G	.	A	.	.	.	.	.	.
HPV58-E7 28S	.	G	.	A	G	G	T	A	T	.
HPV58-E7 48S	A	G	.	A	.	.	.	.	.	.
*Without lesion*										
HPV58-E7 29M	A	G	.	A	.	.	.	.	.	.
HPV58-E7 35P	A	G	.	A	.	.	.	.	.	.
Reference amino acid	G			G	T	T		D		
Amino acid position	41			63	64	74		76		
Amino acid changed	R			D	A	A		E		

Nt: Nucleotide. T: threonine, D: aspartic acid, E: glutamate, R: arginine, G: glycine, and A: alanine. LSIL: low-grade squamous intraepithelial lesion. HSIL: high-grade squamous intraepithelial lesion.

**Table 2 microorganisms-11-01915-t002:** LRT test results for detecting sites under positive (diversifying) selection. Values between parenthesis indicate the posterior probability of the site being positively selected. * indicates *p* > 95%, and ** indicates *p* > 99%.

Model 1	Model 2	LRT	Positively Selected Sites
M0	M3	48.63069	-
M1	M2	31.954069	41G (0.979 *) 63G (1.000 **) 64T (0.826) 74T (0.976 *) 76D (0.997 **)
M1	M3	31.954662	41G (0.997 **) 63G (1.000 **) 64T (0.893) 74T (0.997 **) 76D (1.000 **)
M7	M8	32.308663	41G (0.987 *) 63G (1.000 **) 64T (0.857) 74T (0.985 *) 76D (0.999 **)

**Table 3 microorganisms-11-01915-t003:** Predicted impact of mutations on the structural stability of HPV58 E7 protein, as determined by Site Directed Mutator. Positive values of ΔΔG are associated with increased protein stability. Negative values of ΔΔG are associated with decreased protein stability.

Protein	Mutation	Predicted ΔΔG	Outcome
E7	G41R	−0.14	Reduced stability
	G63D	−2.48	Reduced stability
	T64A	−0.12	Reduced stability
	T74A	1.26	Increased stability
	D76E	1.25	Increased stability

## Data Availability

We shared the data if it request us.

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
