# Peer review of "E7 Oncogene HPV58 Variants Detected in Northeast Brazil: Genetic and Functional Analysis"

_microorganisms, 2023, doi:10.3390/microorganisms11081915_

Round 1

Reviewer 1 Report

Aside from English editing comments, I have a few small questions/suggestions.

1) On lines 245-246, there’s the sentence “A total of two samples positive for HPV58 were excluded due to the absence of a pathological diagnosis.”, what’s the difference between that and the “without lesion” pathological diagnosis?  Can you clarify this a bit in text?

 2) Group samples together by pathological analysis in Table 1

3) Suggestions for Figure 4:  Standardize the size of the figure labels for Figure 4 “A)”, “B)”, “C)”, “D)”.  Also, make the standard deviation bars more visible in Figure 4D.

Minor edits

Line 2: Change “HPV 58” to “HPV58”.

Line 22: Add “the” between “in” and “metropolitan”.

Lines 22-23: Change “The most frequent HPV type was HPV16, 18, and 58, respectively” to “The most frequent HPV types were, in descending order of abundance, HPV16, 18, and 58.”

Lines 26-29: Change “Regarding the interaction of the E7 variant of HPV58 with the signaling of the NF-kB pathway, we observed that the variant HPV58/UFPE-54S decreased the activity of the pathway
when compared to the prototype and the other variants behaved similarly to the prototype and the
prototype increased the activity of the pathway when compared to pcDNA” to “Utilizing a NF-kB reporter assay, we observed a decrease in NK-kB pathway activity with the E7 variant HPV58/UFPE-54S when compared to WT E7 prototype; other detected E7 HPV58 variants had similar NF-kB pathway activity to WT E7 prototype.”

Lines 38-44: Change “Together, they reach the third position in tumor frequency (about 16.340 new cases were reported in 2016) and the fourth in cause of deaths in the female population of this country, constituting a serious public health issue, as it occurs in other developing countries [3]. It was estimated by the World Health Organization (WHO) that 84% of cervical cancer cases occurred in developing countries in 2018, with about 311.000 women deaths. This issue is mainly due to failures in the execution of public health programs for the screening of this cancer [4].” To “Together, they are the 3rd leading cause of carcinogenesis (about 16,340 new cases were reported in 2016) and the 4th leading cause of death in Brazilian woman, constituting a serious public health issue; this carcinogenesis trend is similar to what is being observed in other developing countries. The World Health Organization (WHO) estimated that 84% of cervical cancer cases occurred in developing countries in 2018, leading to about 311,000 deaths in women.  This excess death is thought to be mainly due to failure in the execution of public health programs for cervical cancer screening.”

Line 45: Change “Northeast of Brazil” to “Brazilian Northeast”.

Line 50: Add a space between “E7” and [7].

Line 53: Add commas before “which” and “as”.

Line 60: Make “variant” plural.

Line 68: Change “In unpublished data, our group has found that different variants of E7 oncogene” to “Our group has unpublished data indicating that different variants of the E7 oncogene…”

Line 70: Change “wild gene” to “wild-type E7 gene”.

Line 84: Change “aging from 18 to 77 years old” to “ages ranging from 18 to 77 years old” and “37,5” to “37.5”.

Line 101: Convert the “[Ma et al., 2006]” reference to the proper format.

Lines 108-109: Convert the “[Manos et al.,1989].” reference to the proper format.

Line 140: Add a space between “MEGA11” and “[21]”.

Line 164: Italicize “in silico”.

Line 172: Add a space between “server” and “[27]”.

Line 173: Add a space between “TM-align” and “[31]”.

Lines 200-201: Change “, is was made a co-transfection into keratinocytes resulting from cervical carcinoma (C33A).” to “C33A cells, keratinocytes resulting from cervical carcinoma, were co-transfected with an NF-κB reporter and constructs encoding the HPV58 E7 variants.”

Line 202: Change “5x105” to “5x105”.

Line 207: Remove “even”.

Lines 208-211: “Furthermore, all cell groups were co-transfected with an NF-kB dependent firefly luciferase reporter, which is constructed with three kB factor binding sites being Named (kB) 3-Luc (1 μg) (BCCMTM/LMBP, Gent, Belgium) and a plasmid expressing Renilla luciferase (1 ng) as a luminescence normalizer.”  To “The NF-kB dependent firefly luciferase reporter used for the co-transfections was constructed with three kB factor binding sites named (kB) 3-Luc; 1 µg NF-kB reporter was used per co-transfection, along with 1 ng of a Renilla luciferase-expressing plasmid to normalize luminescence values”.

Line 215: Change “Relative Luciferase unit” to “Relative luciferase units (RLUs)…”.

Lines 223-224: Change “Normalization of gene expression adopted the reference genes GAPDH and ACT.” To “Gene expression was normalized using the reference genes GAPDH and ACT.”

Line 236:  Add comma after “Therefore”.

Lines 336-340: Change “However, when compared to the E7HPV-58 prototype, the 58S and 60M variants were similar NF-kB induction compared to the E7 prototype. with the E7 prototype of HPV-58The studied variants were transfected and their effects on the NF-kB pathway were compared with those found in cells expressing the HPV-58 reference E7 (prototype).” To “However, the 58S and 60M variants showed similar NF-kB induction as compared to the E7 prototype of HPV58.”

Lines 355-357: Change “The reduction in NF-kB activity caused by the E7-54S variant could be due to a low amount of E7 mRNA. In this way, the E7 expression levels of the different variants were evaluated (Figure 4D).” to “We hypothesized that the reduction of NF-kB activity caused by the E7-54S variant could be due to decreased level of E7 mRNA.  Therefore, we evaluated the mRNA levels of the different variants (Figure 4D)”.

Line 367:  Add a space between “immortalization” and “[38]”.

Line 392:  Add a space between “.” and “I”.

Line 402: Rewrite “[52][42,53]” to “[42, 52, 53]” and make “evidences” singular.

Line 406: Rewrite “E7 gene interact” to “E7 gene product interacts…”.

Line 410:  Add a space between “pathway” and “[59,60]”.

Line 414:  Add a space between “signaling” and “[61]”.

Line 418: Change the comma to a semicolon.

Line 423: Change “HPV -58” to “HPV58”.

Line 433: Add a space between “mechanisms” and “[68–70].”.

Line 438: Change “has” to “have” and add a space between “neoplasia” and “[7]”.

Line 443: Change “bellow” to “decreased”.

Author Response

REBUTTAL LETTER

Reviewer 1

Comments and Suggestions for Authors

Aside from English editing comments, I have a few small questions/suggestions.

1) On lines 245-246, there’s the sentence “A total of two samples positive for HPV58 were excluded due to the absence of a pathological diagnosis.”, what’s the difference between that and the “without lesion” pathological diagnosis?  Can you clarify this a bit in text?

Answer: It means to say that these two samples were excluded due to the fact that they did not present conclusive results regarding the cervical cancer screening test. This information was clarified in the text.

 2) Group samples together by pathological analysis in Table 1

Answer: This suggestion was accepted.

3) Suggestions for Figure 4:  Standardize the size of the figure labels for Figure 4 “A)”, “B)”, “C)”, “D)”.  Also, make the standard deviation bars more visible in Figure 4D.

Answer: This suggestion was accepted.

Comments on the Quality of English Language

Answer: These suggestions were accepted.

  • Line 2: Change “HPV 58” to “HPV58”.
  • Line 22: Add “the” between “in” and “metropolitan”.
  • Lines 22-23: Change “The most frequent HPV type was HPV16, 18, and 58, respectively” to “The most frequent HPV types were, in descending order of abundance, HPV16, 18, and 58.”
  • Lines 26-29: Change “Regarding the interaction of the E7 variant of HPV58 with the signaling of the NF-kB pathway, we observed that the variant HPV58/UFPE-54S decreased the activity of the pathway when compared to the prototype and the other variants behaved similarly to the prototype and the prototype increased the activity of the pathway when compared to pcDNA” to “Utilizing a NF-kB reporter assay, we observed a decrease in NK-kB pathway activity with the E7 variant HPV58/UFPE-54S when compared to WT E7 prototype; other detected E7 HPV58 variants had similar NF-kB pathway activity to WT E7 prototype.”
  • Lines 38-44: Change “Together, they reach the third position in tumor frequency (about 16.340 new cases were reported in 2016) and the fourth in cause of deaths in the female population of this country, constituting a serious public health issue, as it occurs in other developing countries [3]. It was estimated by the World Health Organization (WHO) that 84% of cervical cancer cases occurred in developing countries in 2018, with about 311.000 women deaths. This issue is mainly due to failures in the execution of public health programs for the screening of this cancer [4].” To “Together, they are the 3rd leading cause of carcinogenesis (about 16,340 new cases were reported in 2016) and the 4th leading cause of death in Brazilian woman, constituting a serious public health issue; this carcinogenesis trend is similar to what is being observed in other developing countries. The World Health Organization (WHO) estimated that 84% of cervical cancer cases occurred in developing countries in 2018, leading to about 311,000 deaths in women. This excess death is thought to be mainly due to failure in the execution of public health programs for cervical cancer screening.”
  • Line 45: Change “Northeast of Brazil” to “Brazilian Northeast”.
  • Line 50: Add a space between “E7” and [7].
  • Line 53: Add commas before “which” and “as”.
  • Line 60: Make “variant” plural.
  • Line 68: Change “In unpublished data, our group has found that different variants of E7 oncogene” to “Our group has unpublished data indicating that different variants of the E7 oncogene…”
  • Line 70: Change “wild gene” to “wild-type E7 gene”.
  • Line 84: Change “aging from 18 to 77 years old” to “ages ranging from 18 to 77 years old” and “37,5” to “37.5”.
  • Line 101: Convert the “[Ma et al., 2006]” reference to the proper format.
  • Lines 108-109: Convert the “[Manos et al.,1989].” reference to the proper format.
  • Line 140: Add a space between “MEGA11” and “[21]”.
  • Line 164: Italicize “in silico”.
  • Line 172: Add a space between “server” and “[27]”.
  • Line 173: Add a space between “TM-align” and “[31]”.
  • Lines 200-201: Change “, is was made a co-transfection into keratinocytes resulting from cervical carcinoma (C33A).” to “C33A cells, keratinocytes resulting from cervical carcinoma, were co-transfected with an NF-κB reporter and constructs encoding the HPV58 E7 variants.”
  • Line 202: Change “5x105” to “5x105”.
  • Line 207: Remove “even”.
  • Lines 208-211: “Furthermore, all cell groups were co-transfected with an NF-kB dependent firefly luciferase reporter, which is constructed with three kB factor binding sites being Named (kB) 3-Luc (1 μg) (BCCMTM/LMBP, Gent, Belgium) and a plasmid expressing Renilla luciferase (1 ng) as a luminescence normalizer.” To “The NF-kB dependent firefly luciferase reporter used for the co-transfections was constructed with three kB factor binding sites named (kB) 3-Luc; 1 µg NF-kB reporter was used per co-transfection, along with 1 ng of a Renilla luciferase-expressing plasmid to normalize luminescence values”.
  • Line 215: Change “Relative Luciferase unit” to “Relative luciferase units (RLUs)…”.
  • Lines 223-224: Change “Normalization of gene expression adopted the reference genes GAPDH and ACT.” To “Gene expression was normalized using the reference genes GAPDH and ACT.”
  • Line 236: Add comma after “Therefore”.
  • Lines 336-340: Change “However, when compared to the E7HPV-58 prototype, the 58S and 60M variants were similar NF-kB induction compared to the E7 prototype. with the E7 prototype of HPV-58The studied variants were transfected and their effects on the NF-kB pathway were compared with those found in cells expressing the HPV-58 reference E7 (prototype).” To “However, the 58S and 60M variants showed similar NF-kB induction as compared to the E7 prototype of HPV58.”
  • Lines 355-357: Change “The reduction in NF-kB activity caused by the E7-54S variant could be due to a low amount of E7 mRNA. In this way, the E7 expression levels of the different variants were evaluated (Figure 4D).” to “We hypothesized that the reduction of NF-kB activity caused by the E7-54S variant could be due to decreased level of E7 mRNA. Therefore, we evaluated the mRNA levels of the different variants (Figure 4D)”.
  • Line 367: Add a space between “immortalization” and “[38]”.
  • Line 392: Add a space between “.” and “I”.
  • Line 402: Rewrite “[52][42,53]” to “[42, 52, 53]” and make “evidences” singular.
  • Line 406: Rewrite “E7 gene interact” to “E7 gene product interacts…”.
  • Line 410: Add a space between “pathway” and “[59,60]”.
  • Line 414: Add a space between “signaling” and “[61]”.
  • Line 418: Change the comma to a semicolon.
  • Line 423: Change “HPV -58” to “HPV58”.
  • Line 433: Add a space between “mechanisms” and “[68–70].”.
  • Line 438: Change “has” to “have” and add a space between “neoplasia” and “[7]”.
  • Line 443: Change “bellow” to “decreased”.

Reviewer 2 Report

The work addresses an important topic in studying the genetic diversity of HPV underlying different infectious and carcinogenic potential of different lineages. The amount of data on this topic is still limited, and therefore any new information is relevant.

The main problem of the manuscript is a significant number of grammatical errors and stylistic errors that make it difficult to perceive. I strongly recommend using the editorial service.

To better understand the results, the reader could benefit from general information on the distribution of diagnoses in the patient sample analyzed and the correspondence of HPV genotypes to these diagnoses. My attention is drawn to the very high (more than 70%) percentage of HPV-positive tests among the patients enrolled. Was it really a random sample, or was it preselected for some indicators?

The finding that "The E7 oncoprotein from HPV16 reduces NF-kB activity, while the E7 from HPV58 increases it" needs a thorough explanation, since the typical effect of this oncoprotein on NF-kB activity is suppression. It is necessary to unequivocally exclude the possibility of incorrect conclusions related to purely technical biases of research.

Materials and Methods indicated that HPV genotyping was performed using primers specific for 58 genotypes, but it was not clear how genotypes 18 and 31 were determined.

The results given in Tables 1 and 2, in my opinion, could be combined into one table.

In the Discussion of Results section, I would like to see the opinion of the authors regarding the potential use of such data for patient management in connection with the prevention and treatment of cervical neoplasia and cancer.

My overall impression is that the article can be published after some revision and substantial editing.

The main problem of the manuscript is a significant number of grammatical errors and stylistic errors that make it difficult to perceive. I strongly recommend using the editorial service.

Author Response

REBUTTAL LETTER

Reviewer 2

The main problem of the manuscript is a significant number of grammatical errors and stylistic errors that make it difficult to perceive. I strongly recommend using the editorial service.

Answer: Grammatical errors in the manuscript were corrected as requested.

To better understand the results, the reader could benefit from general information on the distribution of diagnoses in the patient sample analyzed and the correspondence of HPV genotypes to these diagnoses. My attention is drawn to the very high (more than 70%) percentage of HPV-positive tests among the patients enrolled. Was it really a random sample, or was it preselected for some indicators?

Answer: Sampling was obtained at random.

The finding that "The E7 oncoprotein from HPV16 reduces NF-kB activity, while the E7 from HPV58 increases it" needs a thorough explanation, since the typical effect of this oncoprotein on NF-kB activity is suppression. It is necessary to unequivocally exclude the possibility of incorrect conclusions related to purely technical biases of research.

Answer: We adjusted the information that was wrong so that it could facilitate the understanding of this aspect, since our work observed that the E7 gene has a different action on the NF-kB pathway between the genotypes of HPV16 and HPV58. Having the potential to modulate infected cells acting differently in the process of cancer development, since both genotypes, regardless of how they are expressed in this pathway, are oncogenic. As exemplified in the text, this phenomenon of differential expression between genotypes and between variants is not limited only to the E7 gene, but can also be observed for the E5 gene of HPV16 and HPV31, according to a study by Silva et al. (2020).

Materials and Methods indicated that HPV genotyping was performed using primers specific for 58 genotypes, but it was not clear how genotypes 18 and 31 were determined.

Answer: The specific primers used for the genotyping of HPV16 and HPV31 were added in the text.

The results given in Tables 1 and 2, in my opinion, could be combined into one table.

Answer: This suggestion was accepted.

In the Discussion of Results section, I would like to see the opinion of the authors regarding the potential use of such data for patient management in connection with the prevention and treatment of cervical neoplasia and cancer.

Answer: We accepted the suggestion and added a paragraph briefly explaining the importance of the

NF-kB pathway in carcinogenesis and its impact on patient clinical outcomes, citing articles of Cai, Het al 2020, Tilborghs S, et al 2017 and Schrank T.P, et al 2022. Furthermore, we included our opinion on how modulation of the pathway by HPV can be utilized in clinical practice.

References:

  • Tilborghs S, Corthouts J, Verhoeven Y, Arias D, Rolfo C, Trinh XB, van Dam PA. The role of Nuclear Factor-kappa B signaling in human cervical cancer. Crit Rev Oncol Hematol. 2017 Dec;120:141-150. doi: 10.1016/j.critrevonc.2017.11.001. Epub 2017 Nov 7. PMID: 29198328.
  • Schrank TP, Prince AC, Sathe T, Wang X, Liu X, Alzhanov DT, Burtness B, Baldwin AS, Yarbrough WG, Issaeva N. NF-κB over-activation portends improved outcomes in HPV-associated head and neck cancer. Oncotarget. 2022 May 24;13:707-722. doi: 10.18632/oncotarget.28232. PMID: 35634245; PMCID: PMC9131933.
  • Cai H, Yan L, Liu N, Xu M, Cai H. IFI16 promotes cervical cancer progression by upregulating PD-L1 in immunomicroenvironment through STING-TBK1-NF-kB pathway. Biomed Pharmacother. 2020 Mar;123:109790. doi: 10.1016/j.biopha.2019.109790. Epub 2019 Dec 30. Erratum in: Biomed Pharmacother. 2020 Jun;126:110077. PMID: 31896065.
